# Identification and assessment of a comprehensive set of structural factors associated with hospital costs in Switzerland

**Michael M. Havranek**[1] *, **Josef Ondrej**[1], **Stella Bollmann**[1], **Philippe K. Widmer**[2], **Simon Spika**[3], **Stefan Boes**[1]

**1** Department of Health Sciences and Medicine, University of Lucerne, Lucerne, Switzerland, **2** Spital Limmattal Zurich, Zurich, Switzerland, **3** University Hospital Zurich, Zurich, Switzerland

* michael.havranek@unilu.ch

**Data Availability Statement:** The data used in this project is owned by a third-party organization called Verein SpitalBenchmark, which collects and confidentially benchmarks financial data among its

## Abstract

Structural factors can influence hospital costs beyond case-mix differences. However, accepted measures on how to distinguish hospitals with regard to cost-related organizational and regional differences are lacking in Switzerland. Therefore, the objective of this study was to identify and assess a comprehensive set of hospital attributes in relation to average case-mix adjusted costs of hospitals. Using detailed hospital and patient-level data enriched with regional information, we derived a list of 23 cost predictors, examined how they are associated with costs, each other, and with different hospital types, and identified principal components within them. Our results showed that attributes describing size, complexity, and teaching-intensity of hospitals (number of beds, discharges, departments, and rate of residents) were positively related to costs and showed the largest values in university (i.e., academic teaching) and central general hospitals. Attributes related to rarity and financial risk of patient mix (ratio of rare DRGs, ratio of children, and expected loss potential based on DRG mix) were positively associated with costs and showed the largest values in children's and university hospitals. Attributes characterizing the provision of essential healthcare functions in the service area (ratio of emergency/ ambulance admissions, admissions during weekends/ nights, and admissions from nursing homes) were positively related to costs and showed the largest values in central and regional general hospitals. Regional attributes describing the location of hospitals in large agglomerations (in contrast to smaller agglomerations and rural areas) were positively associated with costs and showed the largest values in university hospitals. Furthermore, the four principal components identified within the hospital attributes fully explained the observed cost variations across different hospital types. These uncovered relationships may serve as a foundation for objectifying discussions about cost-related heterogeneity in Swiss hospitals and support policymakers to include structural characteristics into cost benchmarking and hospital reimbursement.

hospital members. We had to sign strict confidentiality and data protection agreements to be allowed to work with the data. Consequently, we are not allowed to share a minimal dataset publicly. However, the data we used is also available by the Swiss Federal Office of Statistics (contact via gesundheit@bfs.admin.ch) for researchers who meet the criteria for access to the data.

**Funding:** This study (MMH) was funded by the Research Grant 38988.1 IP-SBM of the Swiss Innovation Agency (Innosuisse, https://www.innosuisse.ch/inno/de/home.html), which promotes science-based innovations commissioned by the Swiss Confederation. The funders had no role in study design, data collection and analysis, decision to publish, or preparation of the manuscript.

**Competing interests:** The authors have declared that no competing interests exist.

## 1. Introduction

Diagnosis-related group (DRG) hospital reimbursement systems classify inpatient cases into cost-homogenous groups based on standardized patient and procedural characteristics [1]. This approach relies on the assumption that after sufficient adjustment for relevant case attributes, expected treatment costs are identical across efficiently operating hospitals. However, previous research has provided ample evidence of structural influences (such as organizational or regional factors) that influence hospital costs beyond case-mix differences of hospitals (see, e.g., [2, 3]). If such structural differences are exogenous (i.e., uncontrollable for hospitals), they need to be included in the reimbursement scheme to ensure fair payments across hospitals facing different circumstances. Because of this, DRG systems of most countries incorporate some organizational or regional factors into their reimbursements. However, which factors are included varies widely between countries [1].

In Switzerland, hospital payments are determined based on relative cost weights informing about the resource requirements of DRGs in relation to the average inpatient treatment costs in the country. The conversion of these relative DRG weights into actual payments is achieved by multiplying them with a base rate. By law, this rate is to be negotiated individually between hospitals and payers (health insurers) in comparison with a national benchmark of what is considered to be an "efficiently operating hospital". The benchmark is established based on a specific percentile (in the distribution) of the average case-mix adjusted costs (calculated as the sum of total costs divided by the sum of cost weights of all patients in a hospital) of all Swiss hospitals.

This procedure potentially allows for the inclusion of justified exogenous organizational or regional differences between hospitals, but there exists no consensus on which structural factors should be considered in benchmarking and reimbursement. Previous research has identified considerable heterogeneity across Swiss hospitals (both between and within different hospital types, see, e.g., [4]) and has discussed certain exogenous structural factors that are related to specific aspects of cost differences between hospitals [2, 3]. Based on that, there have been propositions on how to include exogenous cost-related factors into benchmarking and reimbursement [5, 6]. However, the two suggested approaches rely on very different structural factors. To this date, these factors as well as their operationalizations have never been studied and assessed jointly and systematically. Moreover, many of the previously suggested factors appear highly correlated with others, which presumably leads to issues of multicollinearity that make it difficult to interpret parameters in estimated hospital cost functions and adds complexity without increasing predictive power. Therefore, the aim of this study was to integrate and extend previous findings by systematically identifying and assessing various organizational and regional factors that explain cost variations across Swiss hospitals.

More specifically, our goal was to define a comprehensive set of variables that can be used to distinguish hospitals regarding different cost-related aspects. In order to achieve this, we examined 23 hospital attributes that were hypothesized to exhibit relationships with costs (based on previous literature and expert inputs). We investigated how these attributes are associated with each other and with different hospital types, and we identified four underlying principal components within them. In addition, we tested the practical relevance of our identified factors by showing that they can fully explain the observed cost variations across different hospital types.

From the viewpoint of an international readership, the Swiss DRG system may be of interest because it is an almost fully prospective, DRG-reimbursed hospital financing system with only few special payments [1]. This implies that relationships between structural factors and hospital costs can be observed more directly than may be possible in other settings. However,

this is also precisely the reason why the influence of such factors on hospital costs is a matter of high practical importance in the Swiss setting. Our subsequently reported findings may contribute to objectify present discussions about cost-related heterogeneity in hospitals and provide a basis to incorporate structural differences into benchmarking and reimbursement.

## 2. Methods

### 2.1. Data source

We used data of DRG-reimbursed inpatient stays from 2019 in Switzerland provided by the hospital association "Verein SpitalBenchmark", including:

- routine administrative patient data ("Medizinische Statistik der Krankenhäuser", for further information, available in German/ French/ Italian, see https://www.bfs.admin.ch/bfs/de/home/statistiken/gesundheit/erhebungen/ms.html),

- organizational data on hospitals ("Krankenhausstatistik", for further information, available in German/ French/ Italian, see https://www.bfs.admin.ch/bfs/de/home/statistiken/gesundheit/erhebungen/ks.html),

- specific cost data at the patient level ("Statistik diagnosebezogener Fallkosten", for further information, available in German/ French/ Italian, see https://www.bfs.admin.ch/bfs/de/home/statistiken/gesundheit/erhebungen/fks.html), and

- aggregated cost data at the hospital level ("Kosten- und Leistungsdaten für ITAR_K", for further information, available in German/ French/ Italian, see https://www.hplus.ch/de/rechnungswesen/itar-k).

    In addition:

- publicly available geographic information from the Federal Statistical Office (FSO, "Gemeindetypologie", for further information, available in German/ French/ Italian, see https://www.bfs.admin.ch/bfs/de/home/statistiken/kataloge-datenbanken/karten.assetdetail.2543279.html),

- commercially available socioeconomic information from MicroGIS (https://microgis.ch), and

- commercially available location information of hospitals, nursing and elderly homes, as well as private practices of physicians from Acxiom (www.acxiom.de)

    were linked to patient and hospital data.

    The data contained in total 1,360,975 cases from 120 hospitals, which are about 93% of all stationary cases in Switzerland [7]. Cases that were ongoing (i.e., not concluded, $n = 12,755$), cases with missing DRG codes, cost weights, or costs ($n = 12,795$), cases from a rehabilitation clinic treating only a small number of DRG-reimbursed cases ($n = 319$, as has been proposed previously [5]), and all psychiatric and rehabilitation cases ($n = 142,081$, which are reimbursed outside of the DRG-system) were excluded from analysis. This led to a final sample of 1,193,025 cases from 119 hospitals.

    Patient-level data was used to examine and replicate previous findings of cost associations at the patient level and to build cost-related attributes in the form of aggregated scores at the hospital level (e.g., rate of emergency admissions). All results will subsequently be reported at the hospital level for all included Swiss hospitals on average as well as separately for the following six hospital types to illustrate the heterogeneity of hospitals between and within hospital types:

- university hospitals (*n* = 5, i.e., academic teaching hospitals with an affiliation to a medical school),

- large central general hospitals (*n* = 38),

- medium-sized and small regional general hospitals (*n* = 41),

- birth centers (*n* = 10),

- children's hospitals (*n* = 3), and

- specialty hospitals (*n* = 22, including surgical and other specialty clinics).

## 2.2. Investigated outcome and predictor variables

As outcome variable, we used average case-mix adjusted costs of hospitals (calculated as the sum of total costs divided by the sum of cost weights of all patients in a hospital) without DRG-independently reimbursed costs. This means that we excluded costs for additional services that are reimbursed either by private health insurance or by supplementary reimbursements (which fund certain high-cost treatments).

Cost-explaining predictor variables were initially selected based on reported relationships between organizational and regional factors and hospital costs in the literature and subsequently verified within the regulatory framework of Switzerland via discussions with experts (using focus group meetings and stakeholder consultations). This procedure ensured that the selected variables are relevant and useful to predict hospital costs in the Swiss setting.

We specifically focused on systematic and for hospitals largely uncontrollable influences on costs, where we defined "uncontrollable influences" as exogenous aspects that cannot be changed by hospitals in the short-term without taking drastic measures (such as, for example, completely relocating their facilities). Based on this, we did neither focus on specific aspects (e.g., comparing individual cantons in Switzerland) nor on aspects that are controllable for hospitals (e.g., staffing levels, readmission frequencies, etc.). If multiple variables were possible to describe the same aspect, we used the broadest and most practically relevant one based on expert input (e.g., the number of beds rather than the number of operating rooms to describe the size of a hospital). Furthermore, we chose continuous variables over binary ones, when possible (e.g., the rate of emergency admissions instead of the presence of an emergency room) to make relationships with costs comparable across a spectrum of values for different hospitals, increase precision in estimates due to higher variability in the data, and allow for additional flexibility in functional form. For overview purposes, the reported variables are consecutively numbered (as V1-V23, throughout the article) and are grouped into attributes relating to size and services, patient mix and specialization, and region and location of hospitals (here in Table 1):

**2.2.1. Size and services.** Hospital size and teaching intensity have been found both in Switzerland and internationally to be positively related to hospital costs [5, 6, 8, 9]. Thus, we selected multiple variables describing slightly different aspects of hospitals' size and breadth of provided services (V1-V4) as well as teaching intensity and breadth (V5-V7) to investigate their associations with costs.

**2.2.2. Patient mix and specialization.** Several aspects of patient mix and hospital specialization have previously been reported in association with hospital cost variations [2, 3, 5, 6]. In contrast to previous studies, we aimed to investigate their influence on costs independent of hospital size (as an inherently present confounding factor in many variables). In order to achieve this, we expressed these variables (V8-V17) as ratios or rates (in relation to

**Table 1. Description of predictor variables.**

| Size and services | |
|---|---|
| **Variable** | **Operationalization/ Description** |
| V1 - Number of beds | as indicator for hospital size |
| V2 - Number of discharges (per year) | as indicator for patient volume |
| V3 - Number of departments | e.g., internal medicine, surgery, obstetrics and gynecology, etc., according to the classification of the FSO |
| V4 - Number of types of services | consisting of acute care, psychiatry, and rehabilitation |
| V5 - Number of trained personnel groups | consisting of residents, medical students, and other health professions |
| V6 - Rate of residents to patients | = sum of full-time equivalents of residents / number of discharges |
| V7 - Costs for research and development | DRG-independent costs for R&D in Swiss francs |
| **Patient mix and specialization** | |
| **Variable** | **Operationalization/ Description** |
| V8 - Case-mix index | = sum of cost weights / number of discharges |
| V9 - Ratio of rare DRGs | using 200 cases across all hospitals per year as threshold for rare DRGs |
| V10 - Ratio of deliveries | only counting deliveries without complications or complicating procedures |
| V11 - Ratio of children | including ages 0–17, but excluding healthy newborns delivered without complications or complicating procedures |
| V12 - Ratio of admissions from nursing homes | including admissions from elderly homes, nursing homes, and other sociomedical institutions |
| V13 - Ratio of emergency/ ambulance admissions | excluding deliveries without complications or complicating procedures |
| V14 - Ratio of admissions during weekend/ night | excluding deliveries without complications or complicating procedures |
| V15 - Ratio of admissions from outside of canton | as indicator for hospital specialization |
| V16 - Rate of DRGs to patients | = number of distinct DRGs treated at hospitals / number of discharges |
| V17 - Rate of specialized services to patients | = number of distinct services* requiring equipment or expertise not present in all hospitals / number of patients |
| V18 - Expected loss potential based on DRG mix | as indicator for financial risk taken on by hospitals based on their patient mix, see description in main text |
| **Region and location** | |
| **Variable** | **Operationalization/ Description** |
| V19 - Location in large agglomeration | hospitals located in municipalities within a large agglomeration |
| V20 - Location in medium-sized/ small agglomeration | hospitals located in municipalities within a medium-sized or small agglomeration |
| V21 - Location in peri-urban/ rural area | hospitals located in municipalities within a peri-urban or rural area |
| V22 - Median income of patients | as indicator of patients' socioeconomic status |
| V23 - Healthcare density | composite measure of number of hospitals, nursing and elderly homes, and private practices of physicians in region, see description in main text |

Note. * Specialized services include services requiring an emergency room, intensive care unit, computed tomography or magnetic resonance imaging, dialysis, lithotripsy, and radiotherapy equipment.

all discharges or patients). Likewise, deliveries without complications or complicating procedures were investigated separately in variable V10 and excluded from variables V11, V13, and V14.

Financial risk taken on by hospitals based on patient mix, which has previously been shown not to be sufficiently compensated for in Switzerland [3], was represented by a variable we coined "expected loss potential based on DRG mix" of hospitals (V18). It was generated by the following procedure:

1. For each case across all hospitals, we calculated its *loss* as *loss = actual total costs − reimbursed costs*, where *reimbursed costs* were calculated as *cost weight* x *national base rate*.

2. We set *loss* to zero for cases that have $|loss| < t^* \sigma_{loss}$, where $\sigma_{loss}$ is the estimated standard deviation of *loss*, and $t$ is one of several investigated thresholds ($t = 2, 3,$ or 5, see explanation below).

3. For each DRG, we averaged *loss* across all cases of all hospitals with that DRG (including the ones that were set to zero in step 2).

4. For each case across all hospitals, we defined *expected loss* as average *loss* per DRG from step 3 based on the DRG group the case belongs to.

5. For each hospital, we defined its *expected loss potential based on DRG mix* as the average of *expected loss* from step 4 across all cases of all DRGs in that hospital.

With this approach, we are able to quantify the expected loss potential in a single variable by using probabilities and costs from *all* cases in the country without incorporating (in-)efficiencies of individual hospitals. Notably, all investigated thresholds (see step 2) were able to generate variables that were highly associated with hospital costs. However, we will herein report results for the variable with $t = 5$, because this threshold turned out to be best able to distinguish hospitals with the highest financial risk based on their patient mix.

**2.2.3. Region and location.** Systematic geographic differences, which have been found to be related to hospital costs in other countries [8, 10, 11], were examined using a classification system by the FSO dividing Swiss municipalities into categories based on whether they are located within a large, medium-sized, or small agglomeration, or within a peri-urban or rural area. According to preliminary analyses, the original classification containing nine categories was further aggregated into three categories in V19, V20, and V21. Each hospital (group) with locations in more than one municipality was expressed as the percentage of locations within each of the three variables (ranging from 0 to 1). In addition, to further incorporate information about whether hospitals are located within the center or at the edge of an area, variables were extended to inform about which percentage of the vicinity (10 km radius) around the location of hospitals belonged to the respective category.

Lastly, we hypothesized that the socioeconomic status of hospitals' patients and healthcare provider density within hospitals service areas are associated with hospital costs (e.g., due to different compositions of patient mixes of hospitals or due to supply-side factors indirectly leading to higher admission and/ or readmission rates). Socioeconomic status was inferred from median income of the active population at patients' places of residence (V22), while provider density was expressed as a composite healthcare density measure around patients' places of residence averaged across all patients per hospital (V23). To build the latter, we combined z-standardized and averaged information on the number of hospitals, the number of nursing and elderly homes, and the number of private practices of physicians in a 30 km radius around patients' places of residence.

## 2.3. Statistical analyses

We performed the following statistical analyses in our study: First, associations between the predictor variables and hospital costs and among the predictors were investigated

using Pearson correlations and illustrated as heat maps depicting the relationships among the predictors and between the predictors and hospital types. It must be highlighted that pearson correlations (and illustrations using heat maps) were deliberately chosen instead of multiple regression(s) because our goal was to show the bivariate relationships of the investigated variables with hospital costs and because considerable multicollinearity within many variables (see next step and Fig 1 below) prevented us from estimating multiple regression(s).

Second, a principal component analysis (PCA) was performed to identify underlying components within the predictor variables that showed significant associations with costs (excluding V6 due to missing values in the data provided by the hospitals and using varimax as rotation in PCA) and to recognize collinearities. As a point of reference for our subsequent final analysis, an analysis of variance (ANOVA) was used to examine associations between hospital type and hospital costs. In addition, post hoc independent t-tests were conducted on the basis of significant ANOVA main effects.

Finally, exported scores of our principal components were used in an analysis of covariance (ANCOVA) to examine whether the identified components can explain the associations between hospital types and hospital costs and to rule out that there are additional cost-related aspects (across hospital types), which are not yet captured by our predictor variables.

Analyses were conducted with Python (Version 3.8.8) and R (Version 4.0.2), and results were considered significant if $p < 0.05$. In the case of Pearson correlations and post hoc t-tests, results were considered significant after correction for multiple comparisons using the Hochberg method [12], which is a step-up modification of Bonferroni's method, testing each partition hypothesis using all order statistics by formulating a sequence of critical values based on Simes' inequality.

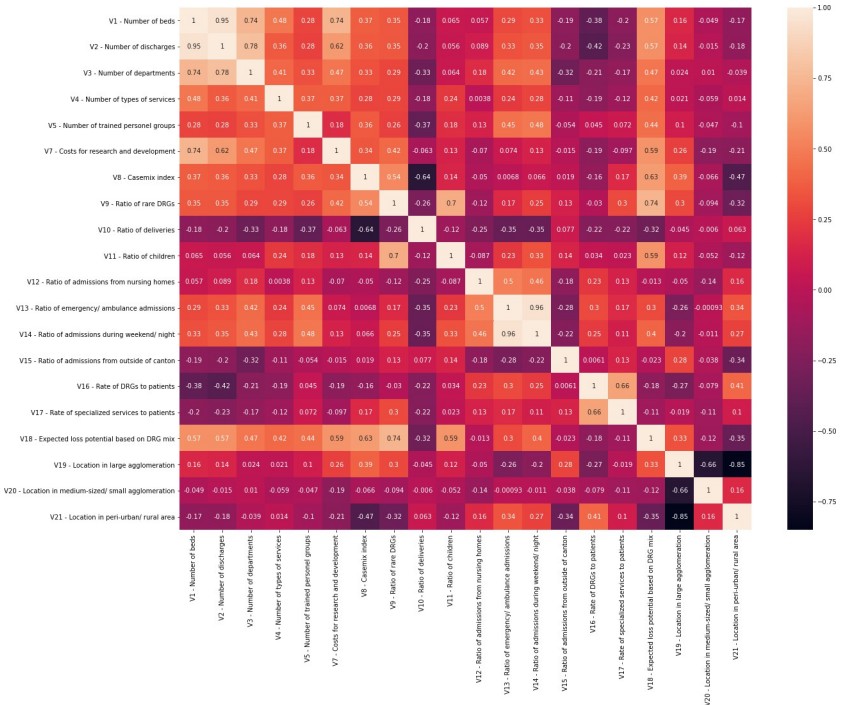

**Fig 1. Heat map of correlation coefficients of variables.** *Note.* Variables with missing values in the data provided by hospitals (V6) and variables not significantly associated with costs (V22 and V23) are excluded.

## 3. Results

As a starting point, descriptive statistics are presented in Table 2 for average case-mix adjusted hospital costs and in Table 3 for a set of selected variables to demonstrate the considerable heterogeneity between hospitals. Furthermore, descriptive statistics for all the investigated variables are provided S1 Table. The presented tables show that Swiss hospitals differ across many important structural dimensions both between and within hospital types (see also statistical analyses in chapter 4.3 below).

### 3.1. Relationships between the predictor variables and hospital costs

Individual relationships between the predictor variables and hospital costs are shown in Table 4. It reveals that most investigated variables exhibit significant associations with costs (with correlation coefficients ranging from -0.284 to 0.451) that generally remain significant after multiple comparison corrections. Exceptions are the number of types of services (V4, $r = 1.161$, $p = 0.081$), and median income of patients (V22) as well as healthcare density (V23), which do not show significant relationships with hospital costs. Most of the variables are positively associated with hospital costs, i.e., they exhibit increasing costs with increasing values in the predictor variables on average. Only variables V10 (ratio of deliveries), V20 (location in medium-sized/ small agglomeration), and V21 (location in peri-urban/ rural area) are negatively associated with costs.

 The following heat maps are used to elucidate the fact that many investigated predictor variables are correlated with each other and show high values in some hospital types. Fig 1 depicts relationships among the investigated variables (with correlation coefficients ranging from -0.85 to 0.96), while Fig 2 informs about the average and z-standardized values across hospital types (ranging from -1.884 to 2.038). Perhaps not surprising, variables describing the size of hospitals (V1-V4) are highly correlated with each other, associated with variables informing about teaching intensity and research and development (V5-V7), and show the largest values in university hospitals and large central hospitals. Similarly unsurprising is the finding that regional factors as described by variables V19-V21 are negatively associated with each other. The highest values of V19 (location within large agglomerations) are observed in university hospitals, the highest values of V20 (location within medium-sized/ small agglomeration) are observed in central and regional general hospitals, and the highest values of V21 (location in peri-urban/ rural areas) are found in regional general hospitals.

 Furthermore, the third set of related attributes can be found in V9, V11, and V18, showing that the ratio of rare DRGs, the ratio of children, and the expected loss potential based on

**Table 2. Means, standard deviations (SD) and ranges of hospital costs (in CHF).**

|               | n   | Mean (SD) Costs | Range Costs     |
|---------------|-----|-----------------|-----------------|
| **University**| 5   | 11,748 (677)    | 11,145–12,905   |
| **Central**   | 38  | 10,393 (541)    | 9,092–11,397    |
| **Regional**  | 41  | 10,518 (773)    | 8,971–12,478    |
| **Birth**     | 10  | 9,525 (893)     | 8,133–11,164    |
| **Children**  | 3   | 11,801 (573)    | 11,249–12,394   |
| **Specialty** | 22  | 10,612 (1,404)  | 8,971–14,400    |
| **All Hospitals** | 119 | 10,496 (960) | 8,133–14,400    |

*Note.* n = sample size, University = university hospitals, Central = central general hospitals, Regional = regional general hospitals, Birth = birth centers, Children = children's hospitals, Specialty = specialty hospitals.

**Table 3. Descriptive statistics (means, standard deviations (SD), and ranges) of selected variables.**

| | *n* | *Mean* (SD) Beds | *Range* Beds | *Mean* (SD) Discharges | *Range* Discharges |
|---|---|---|---|---|---|
| **University** | 5 | 1,143.64 (449.46) | 663.0–1828.73 | 46,544.4 (6,921.79) | 37,503.0–56,157.0 |
| **Central** | 38 | 344.09 (197.12) | 129.56–938.0 | 18,256.05 (9,793.9) | 3,177.0–42,706.0 |
| **Regional** | 41 | 86.39 (54.12) | 7.0–254.47 | 4,867.88 (2,691.18) | 406.0–9,629.0 |
| **Birth** | 10 | 3.61 (1.92) | 1.07–7.34 | 376.5 (286.61) | 110.0–892.0 |
| **Children** | 3 | 136.32 (66.84) | 82.0–210.96 | 6,275.33 (2,205.67) | 4,162.0–8,563.0 |
| **Specialty** | 22 | 62.91 (62.88) | 13.09–301.0 | 2,815.18 (2,103.96) | 153.0–7,798.0 |
| **All Hospitals** | 119 | 203.06 (276.71) | 1.07–1828.73 | 10,172.77 (11,854.06) | 110.0–56,157.0 |
| | *n* | *Mean* (SD) Departments | *Range* Departments | *Mean* (SD) Case-mix | *Range* Case-mix |
| **University** | 5 | 11.6 (1.14) | 10.0–13.0 | 1.44 (0.12) | 1.3–1.59 |
| **Central** | 38 | 6.76 (1.92) | 3.0–11.0 | 0.98 (0.13) | 0.64–1.38 |
| **Regional** | 41 | 4.88 (1.72) | 2.0–9.0 | 0.88 (0.22) | 0.55–1.9 |
| **Birth** | 10 | 1.0 (0.0) | 1.0–1.0 | 0.38 (0.01) | 0.36–0.4 |
| **Children** | 3 | 4.33 (4.16) | 1.0–9.0 | 1.21 (0.23) | 1.06–1.47 |
| **Specialty** | 22 | 2.91 (1.51) | 1.0–6.0 | 1.15 (0.35) | 0.65–2.06 |
| **All Hospitals** | 119 | 5.06 (2.82) | 1.0–13.0 | 0.95 (0.31) | 0.36–2.06 |
| | *n* | *Mean* (SD) Ratio deliveries | *Range* Ratio deliveries | *Mean* (SD) Ratio children | *Range* Ratio children |
| **University** | 5 | 0.1 (0.03) | 0.05–0.14 | 0.08 (0.05) | 0.02–0.13 |
| **Central** | 38 | 0.11 (0.06) | 0.0–0.3 | 0.06 (0.04) | 0.0–0.15 |
| **Regional** | 41 | 0.12 (0.09) | 0.0–0.43 | 0.04 (0.03) | 0.0–0.12 |
| **Birth** | 10 | 0.99 (0.01) | 0.98–1.0 | 0.02 (0.01) | 0.0–0.04 |
| **Children** | 3 | 0.02 (0.01) | 0.02–0.03 | 0.98 (0.01) | 0.97–0.99 |
| **Specialty** | 22 | 0.02 (0.09) | 0.0–0.42 | 0.03 (0.05) | 0.0–0.22 |
| **All Hospitals** | 119 | 0.17 (0.26) | 0.0–1.0 | 0.07 (0.15) | 0.0–0.99 |

*Note. n* = sample size, University = university hospitals, Central = central general hospitals, Regional = regional general hospitals, Birth = birth centers, Children = children's hospitals, Specialty = specialty hospitals.

DRG mix are highly associated with each other and display the largest values in children's hospitals and in the case of the expected loss potential also in university hospitals. Finally, the fourth group of related variables can be observed in V12-V14 (i.e., ratio of admissions from nursing homes, ratio of emergency/ ambulance admissions, and ratio of admission during weekends/ nights), which are correlated highly with each other and show relatively larger values in central and regional general hospitals.

## 3.2. Principal components within the predictor variables

As a next step, the identified principal components within the investigated variables are reported. Parallel analysis (using the psych package in R) comparing the eigenvalues of factors/ components of the observed data with that of random data of the same size suggested four principal components for the investigated variables associated with hospital costs. Table 5 presents the component loadings of the variables (only displaying loadings above 0.3 and below -0.3, respectively).

The first component (C1 in Table 5) reveals high positive loadings from variables describing size, complexity, and teaching intensity (V1-V8), and high negative loadings from variables that were by definition adjusted for hospital size (V16-V17). The second component (C2 in Table 5) displays high loadings from variables describing patient mixes containing rare DRGs (V9), children (V11), and financially risky (V18) cases. The third component (C3 in Table 5)

**Table 4. Pearson correlations (incl. sample sizes (n), means, and standard deviations (SD)) of predictor variables with hospital costs.**

|  | *n* | *Mean (SD)* | *r* | *p* | HB |  |
|---|---|---|---|---|---|---|
| **V1 - Number of beds** | 119 | 203.06 (276.71) | 0.256 | 0.005 | 0.026 | ** |
| **V2 - Number of discharges** | 119 | 10,172.77 (11,854.06) | 0.185 | 0.044 | 0.043 | (*) |
| **V3 - Number of departments** | 119 | 5.06 (2.82) | 0.216 | 0.018 | 0.030 | * |
| **V4 - Number of types of services** | 119 | 1.39 (0.61) | 0.161 | 0.081 | 0.046 | (') |
| **V5 - Number of trained personnel groups** | 119 | 2.17 (1.04) | 0.292 | 0.001 | 0.013 | ** |
| **V6 - Rate of residents to patients** | 107 | 0.25 (0.23) | 0.224 | 0.021 | 0.037 | * |
| **V7 - Costs for research and development** | 119 | 5.16e6 (2.25e7) | 0.274 | 0.003 | 0.020 | ** |
| **V8 - Case-mix index** | 119 | 0.95 (0.31) | 0.342 | 0.000 | 0.007 | *** |
| **V9 - Ratio of rare DRGs** | 119 | 0.02 (0.03) | 0.451 | 0.000 | 0.002 | *** |
| **V10 - Ratio of deliveries** | 119 | 0.17 (0.26) | -0.283 | 0.002 | 0.017 | ** |
| **V11 - Ratio of children** | 119 | 0.07 (0.15) | 0.271 | 0.003 | 0.022 | ** |
| **V12 - Ratio of admissions from nursing homes** | 119 | 0.02 (0.02) | 0.187 | 0.042 | 0.041 | (*) |
| **V13 - Ratio of emergency/ ambul. admissions** | 119 | 0.38 (0.26) | 0.208 | 0.023 | 0.039 | * |
| **V14 - Ratio of admissions on weekends/ nights** | 119 | 0.15 (0.1) | 0.213 | 0.020 | 0.035 | * |
| **V15 - Ratio of admissions outside of canton** | 119 | 0.22 (0.2) | 0.247 | 0.007 | 0.024 | ** |
| **V16 - Rate of DRGs to patients** | 119 | 0.06 (0.05) | 0.216 | 0.019 | 0.033 | * |
| **V17 - Rate of specialized services to patients** | 119 | 0.07 (0.12) | 0.407 | 0.000 | 0.004 | *** |
| **V18 - Expected loss potential based on DRGs** | 119 | 91.47 (89.74) | 0.341 | 0.000 | 0.009 | *** |
| **V19 - Location in large agglomeration** | 119 | 0.24 (0.33) | 0.325 | 0.000 | 0.011 | *** |
| **V20 - Location in med.s./ small agglomeration** | 119 | 0.17 (0.17) | -0.284 | 0.002 | 0.015 | ** |
| **V21 - Location in peri-urban/ rural area** | 119 | 0.59 (0.25) | -0.228 | 0.013 | 0.028 | * |
| **V22 - Median income of patients** | 118 | 65.1 (10.76) | 0.105 | 0.258 | 0.050 |  |
| **V23 - Healthcare density** | 118 | -0.01 (0.96) | 0.115 | 0.213 | 0.048 |  |

*Note*. Variables V6 and V17 were rescaled to make their values readable with the chosen scale. V6 contains missing values because not all hospitals provided information on their number of residents. *r* = correlation coefficient, *p* = p-value, HB = required p-value for significance after multiple comparison correction with the Hochberg method, * = p-value below 0.05,

** = p-value below 0.01,

*** = p-value below 0.001, (*) = p-value below 0.05 but not below HB threshold, (') = p-value below 0.1 but not below HB threshold, ambul. = ambulance, med.s. = medium-sized.

shows high loadings from variables describing the provision of certain essential healthcare functions consisting of the treatment of elderly (V12), emergency (V13), and unscheduled (V14) patients as well as the provision of treatments for broad DRG mixes (V16) and multiple specialized services in relation to patient volume (V17). The fourth and final component (C4 in Table 5) displays high positive and negative loadings from the three regional variables (V19-V21).

### 3.3. Explanation of the association between hospital types and costs

The last part of our analyses served to examine whether the principal components identified within our predictor variables can explain the associations between hospital types and costs. ANOVA results of the association of the categorical variable "hospital type" and hospital costs (*F(df)* = 6.09, *p* = 0.0) show that hospital type is a highly significant predictor of variations in hospital costs. Post hoc independent t-tests presented in Table 6 furthermore reveal that the largest average costs are found in university hospitals (*t* = 3.397, *p* = 0.001) and children's hospitals (*t* = 2.807, *p* = 0.008), whereas the lowest average costs are observed in birth centers (*t* = -3.537, *p* = 0.001). Large central general hospitals and specialty hospitals did not display

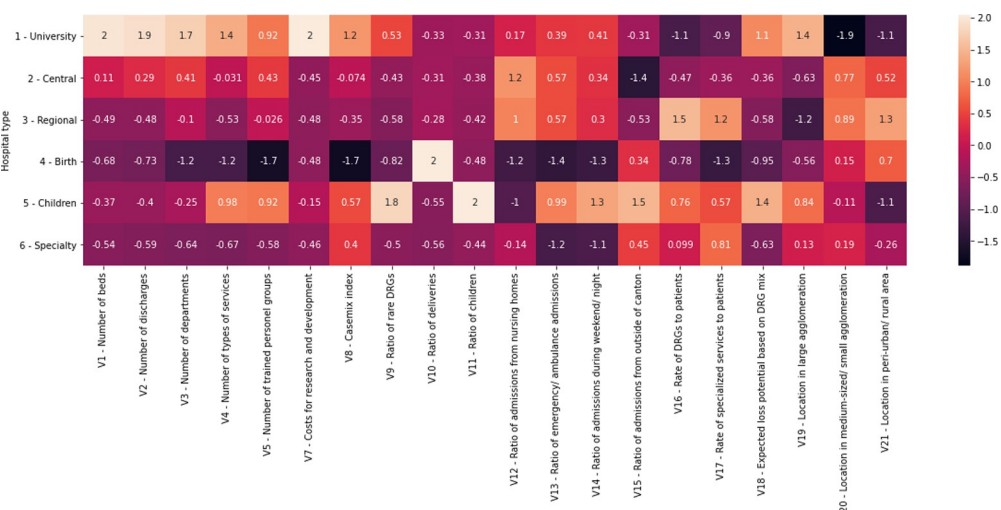

**Fig 2. Heat map of average z-standardized values of variables for different hospital types.** *Note.*
University = university hospitals, Central = central general hospitals, Regional = regional general hospitals,
Birth = birth centers, Children = children's hospitals, Specialty = specialty hospitals. Variables with missing values (V6)
and variables not significantly associated with costs (V22 and V23) are excluded.

significant differences in average costs compared to regional hospitals. However, Table 6 also
reveals that there exists large cost variation within the group of specialty hospitals. Finally,

**Table 5. Loadings of variables on the four identified principal components (C1-C4).**

|  | C1 | C3 | C2 | C4 |
|---|---|---|---|---|
| **V1 - Number of beds** | 0.91 | | | |
| **V2 - Number of discharges** | 0.90 | | | |
| **V3 - Number of departments** | 0.79 | 0.32 | | |
| **V4 - Number of types of services** | 0.52 | | | |
| **V5 - Number of trained personnel groups** | | 0.46 | 0.31 | |
| **V7 - Costs for research and development** | 0.65 | | | |
| **V8 - Case-mix index** | 0.31 | | 0.47 | 0.49 |
| **V9 - Ratio of rare DRGs** | | | 0.85 | |
| **V10 - Ratio of deliveries** | | -0.57 | | |
| **V11 - Ratio of children** | | | 0.82 | |
| **V12 - Ratio of admissions from nursing homes** | | 0.68 | -0.34 | |
| **V13 - Ratio of emergency/ ambulance admissions** | | 0.83 | | |
| **V14 - Ratio of admissions during weekend/ night** | | 0.78 | | |
| **V15 - Ratio of admissions from outside of canton** | -0.34 | | 0.31 | |
| **V16 - Rate of DRGs to patients** | -0.59 | 0.59 | | |
| **V17 - Rate of specialized services to patients** | -0.50 | 0.48 | | |
| **V18 - Expected loss potential based on DRG mix** | 0.55 | | 0.68 | |
| **V19 - Location in large agglomeration** | | | | 0.93 |
| **V20 - Location in medium-sized/ small agglomeration** | | | | -0.67 |
| **V21 - Location in peri-urban/ rural area** | | 0.32 | | -0.75 |

*Note.* Variables with missing values in the data provided by hospitals (V6) and variables not significantly associated
with costs (V22, and V23) are excluded. Only loadings above 0.3 and below -0.3, respectively, are displayed. Varimax
rotation was used in PCA.

Table 6. Means, standard deviations (SD) of hospital costs (in CHF), and results of independent t-tests across different hospital types.

|  | Mean (SD) | t | p | HB |  |
|---|---|---|---|---|---|
| University | 11,748 (677) | 3.397 | 0.001 | 0.020 | ** |
| Central | 10,393 (541) | -0.825 | 0.412 | 0.040 |  |
| Regional | 10,518 (773) | - | - |  |  |
| Birth | 9,525 (893) | -3.537 | 0.001 | 0.010 | ** |
| Children | 11,801 (573) | 2.807 | 0.008 | 0.030 | ** |
| Specialty | 10,612 (1,404) | 0.343 | 0.733 | 0.050 |  |

*Note. t* = t-value, *p* = p-value, HB = required p-value for significance after multiple comparison correction with the Hochberg method, ** = p-value below 0.01, University = university hospitals, Central = central general hospitals, Regional = regional general hospitals, Birth = birth centers, Children = children's hospitals, Specialty = specialty hospitals. T-tests compared each hospital type with regional general hospitals as reference category (-).

ANCOVA results reported in Table 7 show that scores of the four selected components fully explain the associations between hospital type and hospital costs, because hospital type is no longer significantly related to costs after inclusion of the component scores. Among the components, C3 and C4 show the highest explained variation in hospital costs, followed by C1, while C2 has the lowest contribution to the explained variation (and is not statistically significant).

## 4. Discussion

Previous research has shown that structural characteristics can impact hospital costs despite careful adjustment for case-mix differences [2, 3, 5, 6]. Case-mix adjustment of hospital costs produces average costs, which are corrected for case-related characteristics that are incorporated in DRG reimbursements. If remaining differences in case-mix adjusted hospital costs can be explained by exogenous (i.e., for the hospitals uncontrollable) factors, it means that these exogenous factors are not or cannot sufficiently be compensated by the DRG system. Previous studies have looked at selected exogenous factors in isolation, however, without a comprehensive analysis of all cost-related factors beyond case-mix. To the best of our knowledge, this is the first study that integrates and extends these previous findings by identifying and assessing a comprehensive set of organizational and regional variables. The herein identified and subsequently discussed variables explain systematic cost variations and may be used to differentiate hospitals with regard to cost benchmarking and reimbursement.

Table 7. ANCOVA results of the associations between hospital type and the four extracted principal component scores (C1-C4) and hospital costs.

|  | SS | df | F | p |
|---|---|---|---|---|
| Hospital type | 2,129,981 | 5 | 0.714 | 0.614 |
| C1 | 2,560,933 | 1 | 4.294 | 0.041 |
| C2 | 1,495,194 | 1 | 2.507 | 0.116 |
| C3 | 7,608,022 | 1 | 12.756 | 0.001 |
| C4 | 5,463,423 | 1 | 9.160 | 0.003 |
| Residual | 65,012,358 | 109 | - | - |

*Note. SS* = Sum of squares, *df* = degrees of freedom, *F* = F-value, *p* = p-value, hospital type is divided into university hospitals, central general hospitals, regional general hospitals, birth centers, children's hospitals, and specialty hospitals.

## 4.1 Identified predictor variables and underlying principal components

In accordance with previous research [8, 9], we found that several attributes related to size and complexity of hospitals were positively associated with costs, including number of beds (V1), number of discharges (V2), number of departments (V3), and number of types of services (V4, although V4 only displayed a non-significant trend). In our opinion, the reason why these variables and their underlying component are linked to higher costs is related to the complexity that they describe, rather than the size of hospitals, because results from preliminary analyses (not shown) revealed that within a certain department, increasing patient volume is actually associated with lower costs. This finding is consistent with the economies of scale hypothesis, which (among other things) explains why higher patient volume has repeatedly been related to better quality outcomes (see e.g., [13]).

Teaching intensity in the form of the number of trained personnel groups (V5) and rate of residents to patients (V6) as well as research and development (V7) showed positive associations with costs, which is in line with previous findings from other countries [11, 14]. These variables were correlated with the size and complexity of hospitals and showed the largest values in university and large central hospitals. Conversely, the rate of DRGs to patients and the rate of specialized services to patients were inherently negatively correlated with size-related variables and exhibited the largest values in regional hospitals.

As a second set of attributes, we identified the ratio of rare DRGs (V9), the ratio of children (V11), and the expected loss potential based on DRG mix (V19), which were positively associated with costs, highly correlated among each other, and displayed the largest values in children's hospitals, and in the case of the expected loss potential also in university hospitals. This is consistent with previous research that showed that children's hospitals treat a large proportion of rare conditions with a large percentage of high-cost outliers [2, 15–17], while the latter has also been described for university hospitals [3]. The principal component underlying these variables appears to describe the rarity of conditions that certain hospitals treat, which has been argued to require specialized skills and/ or equipment, for which higher wages and upfront capital investments are not sufficiently compensated by the Swiss DRG system. Furthermore, rare DRGs also pose a practical, statistical problem for a DRG system in general because their small sample sizes make accurate prospective cost modeling difficult, which presumably explains the association of V9 (and indirectly also V11) with the financial risk variable V19 (which was designed to characterize this exact problem) [2, 3].

A third cluster of related attributes consisted of the ratio of admissions from nursing homes (V12), the ratio of emergency/ ambulance admissions (V13), and the ratio of admission during weekends/ nights (V14). In congruence with previous international research [18–20], these variables were positively related to hospital costs. In our opinion, this component (which also include V16 and V17) seems to summarize the provision of essential healthcare functions that certain hospitals perform in their service area, consisting of emergency and first point of care functions as well as treatment of a broad DRG mix and provision of multiple specialized services. It may be argued that the provision of these services increases costs because they require additional investments by the hospitals and maintenance of sufficient reserve capacity, which may not be sufficiently compensated by the Swiss DRG system. More precisely, the compensations for these services are not specifically enough directed towards the hospitals that predominantly provide them in their service area.

The regional variables informing about the location of hospitals could be interpreted as a fourth group of related variables. Hospitals with locations within large agglomerations have been found to be positively associated with costs, whereas locations within medium-sized/ small agglomerations or within peri-urban/ rural areas have been found to be negatively

associated with costs. These findings are consistent with studies from other countries comparing hospitals located in urban vs. rural areas, which showed that hospitals in urban areas exhibited higher costs [8, 10, 11]. It has been argued that such regional cost variations arise from differences in salary levels, purchasing power, and real estate costs.

Finally, the only three remaining variables (not discussed yet) are V8 (case-mix index), V10 (ratio of deliveries), and V15 (ratio of admission from outside of canton). The case-mix index showed a positive association with costs, while the ratio of deliveries showed a negative relationship with costs, and the two variables were inversely associated with each other. These findings can be interpreted with respect to previous studies that have shown higher costs in hospitals with higher case-mix indices [6] and other research showing lower costs in birth centers [21, 22]. As in the case of rare disorders discussed above, it can be argued that the treatment of more severe cases requires specialized knowledge and skills, for which higher required salary levels and upfront capital investments are not sufficiently compensated by the Swiss DRG system. The reasoning behind this argument becomes particularly apparent in the opposite case of birth centers, which can specialize themselves in the efficient delivery of a single low-risk service that does not need any reserve capacity because cases with complications are simply referred to the nearest hospital.

The ratio of admissions from outside of the canton of hospitals was designed to assess the association between hospital costs and the specialization of hospitals as indicated by the proportion of patients arriving from outside of the canton to make use of the specialized services that hospitals provide. This reasoning is based on the fact that previous studies [23] have shown higher costs in *certain* specialty hospitals and the finding that this variable has proven to be the best variable to distinguish hospitals within the highly diverse group of specialty clinics. It may be argued that some specialty clinics can specialize in relatively high-volume (low risk) procedures directed towards a large enough pool of patients (i.e., receive enough patients from within their own canton). In contrast, other specialty clinics focusing on low-volume (higher risk) procedures must expand their service area to include patients from other cantons. This latter group (for which the ratio of admissions from outside of the canton variable was designed) may not have a large enough volume of patients, over which they can distribute their additional investments in skills and equipment required to deliver their specialized services. However, this is our interpretation of the empirically observed finding and it cannot be ruled out that other reasons (e.g., the location of hospitals nearby the border of the canton) are also involved in the observed association between hospital costs and the ratio of admissions from outside of the canton.

## 4.2. Practical relevance of the findings

Almost a decade after the Swiss DRG system has been introduced in Switzerland, there still remain large differences in average costs between hospitals, even after adjustment for case-mix [7]. As we have confirmed in this study, one important factor explaining these differences in costs is the hospital type. Among different hospital types, university hospitals (i.e., academic teaching hospitals with an affiliation to a medical school) as well as children's hospitals exhibited higher adjusted average costs than other hospitals, whereas birth centers (focusing exclusively on deliveries without complications or complicating procedures) showed lowest average costs.

In addition, our results revealed that the four principal components extracted from our predictor variables were able to fully explain the associations between hospital types and costs. In our opinion, this shows that we were able to capture the most important differences among hospitals with the presented variables. Furthermore, many investigated variables were

designed to portray the underlying reasons that lie at the heart of cost differences between hospital types and may thus be seen as more "pure" predictors of hospital cost variations rather than hospital type. One example for this is illustrated by the variable expected loss potential based on DRG mix (V18), which elucidate the previously established reasons leading to higher costs in children's and university hospitals (see [2, 3]). However, it is important to keep in mind that while certain variables have specifically been designed to describe characteristic aspects of specific hospital types (such as the ratio of deliveries for birth centers), other variables like the location variables are specifically designed to further distinguish hospitals within the same hospital type. Thus, our presented variables go beyond just explaining common differences that are already found in between-comparisons of hospital types, but also allow us to better explain hospital costs in within-comparisons of hospitals of a specific type.

As discussed in the introduction, the established regulations in Switzerland, dictating that base rates of hospitals are to be negotiated between hospitals and health insurers, allow for the inclusion of justified organizational or regional factors into reimbursements. However, a consensus on which structural factors to consider has previously been lacking. We hope that the present study promotes more empirically founded discussions in the future. In fact, the identified variables and relationships could be used to develop fully empirically based econometric models using exogenous factors to explain hospital costs. Such models could be applied to distinguish legitimate cost differences among hospitals from inefficient hospital operations and to incorporate organizational and regional hospital differences into cost benchmarking and hospital payments in a data-driven way.

Regarding this proposition, a practically relevant finding from this study is the fact that many of the above presented variables describe comparable aspects and are thus highly correlated with each other. On the one hand, this means that some variables could be used more or less interchangeably in econometric modelling. On the other hand, combining such highly related variables in the same econometric model should be avoided to prevent issues of multicollinearity. Consequently, the relationships among the variables presented herein may be used as a reference to select suitable variable combinations to explain different structural aspects related to cost variations.

As our present study focused on identifying and comparing general attributes related to hospital costs to define a set of variables as a basis to distinguish hospitals, one limitation of the study is that we did not further investigate the reasons behind some of the associations that we discovered. This may be an interesting topic for future research. Furthermore, another limitation may be our focus on the bivariate relationships of individual variables with hospitals costs. Our goal was to identify and discuss various separate influences on costs to lay a foundation for policymakers aiming to include some of these factors into cost benchmarking and reimbursement in Switzerland. However, the next step must now lie in determining how the identified structural differences among hospitals could be combined to compensate hospitals for uncontrollable organizational and regional differences. For this, we suggest a regression-based approach, which selects a limited number of independent variables by building on the findings presented herein.

In conclusion, the present study identified and assessed a broad set of hospital attributes that showed significant associations with variations in average case-mix adjusted hospital costs. In addition, the discovered attributes have been described in relation to each other and in relation to different hospital types to highlight potential collinearity issues. Finally, the four underlying components within these variables have been identified, discussed, and shown to fully explain cost differences between hospital types. These findings can be used to objectify discussions about cost-related heterogeneity between hospitals in the future and as a foundation for policymakers to empirically incorporate structural differences into the reimbursement structure in Switzerland.

## Supporting information

**S1 Table. Descriptive statistics (means, standard deviations (SD), and ranges) of all investigated variables.**
(DOCX)

## Acknowledgments

The authors thank the members of the board of directors of "Verein SpitalBenchmark" and in particular its director, Ernst Frank, as well as Fabian Gfeller from SwissDRG for various critical discussions and valuable suggestions.

## Author Contributions

**Conceptualization:** Michael M. Havranek, Philippe K. Widmer, Simon Spika, Stefan Boes.

**Data curation:** Michael M. Havranek, Josef Ondrej, Stella Bollmann.

**Formal analysis:** Michael M. Havranek, Josef Ondrej, Stella Bollmann.

**Funding acquisition:** Michael M. Havranek.

**Investigation:** Michael M. Havranek.

**Methodology:** Michael M. Havranek, Josef Ondrej, Philippe K. Widmer, Simon Spika, Stefan Boes.

**Project administration:** Michael M. Havranek.

**Supervision:** Michael M. Havranek.

**Writing – original draft:** Michael M. Havranek.

**Writing – review & editing:** Michael M. Havranek, Josef Ondrej, Stella Bollmann, Philippe K. Widmer, Simon Spika, Stefan Boes.

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
