## [Decision Letter · Decision Letter 0]

9 Nov 2021

PONE-D-21-25093

Identification and assessment of a comprehensive set of structural factors associated with hospital costs in Switzerland

PLOS ONE

Dear Dr. Havranek,

Thank you for submitting your manuscript to PLOS ONE. After careful consideration, we feel that it has merit but does not fully meet PLOS ONE’s publication criteria as it currently stands. Therefore, we invite you to submit a revised version of the manuscript that addresses the points raised during the review process.

While revising the paper, my recommendation is to follow closely the suggestions of both referees. One report provides rather detailed recommendations concerning the presentation of the material and suggesting a more precise phrasing of several paragraphs.  They are mostly minor points, although some more effort to improve the interpretation of the results is required as well. The other report includes more fundamental critiques concerning  the empirical strategy adopted in the analysis. In this respect, I expect the authors to strengthen the discussion concerning the motivation of their work and of the way the empirical study has been conducted.

We look forward to receiving your revised manuscript.

Kind regards,

Matteo Lippi Bruni, PhD

Academic Editor

PLOS ONE

https://journals.plos.org/plosone/s/file?id=wjVg/PLOSOne_formatting_sample_main_body.pdf and https://journals.plos.org/plosone/s/file?id=ba62/PLOSOne_formatting_sample_title_authors_affiliations.pdf2.

Reviewers' comments:

Reviewer's Responses to Questions

**Comments to the Author**

1. Is the manuscript technically sound, and do the data support the conclusions?

Reviewer #1: Yes

Reviewer #2: Yes

2. Has the statistical analysis been performed appropriately and rigorously? 

Reviewer #1: Yes

Reviewer #2: Yes

3. Have the authors made all data underlying the findings in their manuscript fully available?

Reviewer #1: No

Reviewer #2: No

4. Is the manuscript presented in an intelligible fashion and written in standard English?

Reviewer #1: Yes

Reviewer #2: Yes

5. Review Comments to the Author

Reviewer #1: The authors consider the relationship between case-mix adjusted cost and a range of hospital and regional characteristics in Switzerland. The analysis includes a principal component analysis, in recognition of the high degree of collinearity between groups of potentially relevant variables. The manuscript is presented in a convincing manner and represents a useful contribution. I recommend publication subject to some minor presentational changes.

1. When the authors refer to numbers larger than 1,000 in the text, they use a superscript comma “’” to separate thousands. A subscript comma “,” is more usual.

2. When referring to the unit of analysis, the authors use “on the hospital” or “on the patient” level. “at the hospital level” is what I think is meant.

3. In discussing descriptive results, the authors list mean, standard deviation and range of costs per patient as part of text. This information would be easier to read in Table 2, where a set of other variables are already provided.

4. In Table 2, the authors present a single row for each type of hospital. An additional row, containing the same information across all hospitals together would also be useful. Further, it would add to transparency if equivalent information to that presented in Table 2 was made available for all variables. An appendix or supplementary material would be sufficient for this.

5. In Table 3, V6 (Rate of residents to patients) could be made more informative by rescaling. It should also be noted why some observations are missing for this variable.

6. The two heat maps are difficult to read, most likely due to low resolution.

7. In the Discussion section, case-mix adjustment is referred to as complete. A less absolute term such as considerable would be more defendable.

8. At several points in the Discussion, the authors refer to the issue of up front costs related to skills. It is not clear why differences in the skills of staff would not be covered by wages and so are not up front but regular expenditures. A distinction should be drawn between this and capital investments, which do represent an upfront cost that regular DRG payments don’t capture well.

9. Broadly the authors draw a connection between high costs and insufficient compensation from the DRG system. I agree with this conclusion as an interpretation of the results but it would be useful to remove potential ambiguity by arguing that higher case-mix adjusted costs reflect insufficient compensation nfrom the DRG system.

Reviewer #2: The Authors present a study to integrate and extend previous findings by identifying and assessing various organizational and regional factors that explain systematic cost variations across Swiss hospitals. The aim of the paper is to define a comprehensive set of variables that can be used to distinguish hospitals regarding different cost-related aspects.

The Authors claim that to achieve this, they examine 23 hospital indicators and investigated how these indicators are associated with each other and with different hospital types.

I find the idea interesting but I am not convinced by this work. In fact, I am rather surprised by the methodology used: with such data, the Authors could have estimates a causal relationship between the indicators and the hospital costs, instead of performing Pearson correlations, a principal component analysis (PCA) and an analysis of variance (ANOVA).

Another relevant issue is related to policy implications. From a policy maker's perspective, why is it be relevant to know the effects of 23 cost predictors on hospital costs?

Last, section 5 "discussion" is mostly a repetition of section 4 "results".

6. PLOS authors have the option to publish the peer review history of their article (what does this mean?). If published, this will include your full peer review and any attached files.

Reviewer #1: **Yes: **James Gaughan

Reviewer #2: No

---

## [Author Response · Author response to Decision Letter 0]

31 Jan 2022

Response to each point raised by the reviewers

Reviewer # 1:

1. When the authors refer to numbers larger than 1,000 in the text, they use a superscript comma “’” to separate thousands. A subscript comma “,” is more usual.

Thanks for this suggestion, we have replaced “’” by “,” throughout the text.

2. When referring to the unit of analysis, the authors use “on the hospital” or “on the patient” level. “at the hospital level” is what I think is meant.

Yes, you are right, this has been adapted as well.

3. In discussing descriptive results, the authors list mean, standard deviation and range of costs per patient as part of text. This information would be easier to read in Table 2, where a set of other variables are already provided.

Good point, we have now added a Table 2a depicting mean, standard deviation, and range of costs that were previously included in the main text.

4. In Table 2, the authors present a single row for each type of hospital. An additional row, containing the same information across all hospitals together would also be useful. Further, it would add to transparency if equivalent information to that presented in Table 2 was made available for all variables. An appendix or supplementary material would be sufficient for this.

We agree with both points. An additional row informing about aggregate values across all hospitals is part of Table 2b now and an additional table informing about the other variables has been added to the appendix.

5. In Table 3, V6 (Rate of residents to patients) could be made more informative by rescaling. It should also be noted why some observations are missing for this variable.

Thanks, the rescaling has been done and the information about the reason for the missing values has been provided. 

6. The two heat maps are difficult to read, most likely due to low resolution.

The low resolution of the heat maps seems to occur when the PDF is produced for review in the submission process of PLOS. Both figures actually have a much higher resolution and have passed the PACE digital diagnostic requirements. We will make sure that this issue is resolved together with PLOS before publication. Thanks for pointing it out. 

7. In the Discussion section, case-mix adjustment is referred to as complete. A less absolute term such as considerable would be more defendable.

Thanks, this has been changed.

8. At several points in the Discussion, the authors refer to the issue of upfront costs related to skills. It is not clear why differences in the skills of staff would not be covered by wages and so are not up front but regular expenditures. A distinction should be drawn between this and capital investments, which do represent an upfront cost that regular DRG payments don’t capture well.

Yes, you are right, we have divided these two separate aspects into “salary levels” and “upfront capital investments” now, as you have proposed. However, it must be noted that both aspects are currently not sufficiently compensated by the Swiss DRG system. Because of this, we included both aspects in our discussion.

9. Broadly the authors draw a connection between high costs and insufficient compensation from the DRG system. I agree with this conclusion as an interpretation of the results, but it would be useful to remove potential ambiguity by arguing that higher case-mix adjusted costs reflect insufficient compensation from the DRG system.

That’s a good idea, we have added this to the beginning of the discussion section.

Reviewer # 2:

1a. I am rather surprised by the methodology used: with such data, the Authors could have estimated a causal relationship between the indicators and the hospital costs, instead of performing Pearson correlations, a principal component analysis (PCA) and an analysis of variance (ANOVA).

1b. From a policy maker's perspective, why is it relevant to know the effects of 23 cost predictors on hospital costs?

Thanks for your suggestions and your question, we understand your perspective and would like to give you some more background on why we chose our methodology:

Previous research in Switzerland has only discussed certain specific structural factors that are related to cost differences between hospitals separately. Based on that, there have been propositions for exogenous factors that should be included in cost benchmarking and reimbursement (see references below). However, these propositions lacked transparency on why they selected exactly those specific factors for inclusion and on what other relevant factors would have existed. Thus, we aimed to identify and assess a broad set of exogenous cost-related variables in a comprehensive and transparent manner. 

Our results shall serve as a foundation for objectifying future discussions in Switzerland about which factors exist and which of these should be included in cost benchmarking and reimbursement. The presented relationships shall inform policy-makers about which structural influences on hospitals are not sufficiently compensated by the Swiss DRG system. Furthermore, it shall support them in selecting and combining several specific variables to systematically compensate hospitals for these uncontrollable organizational and regional influences.

Now regarding the estimation of causal relationships between the indicators and the hospitals costs:

We assume you are referring to regression analyses, right? On the one hand, we did not present regressions because this was not the focus of our paper based on the motivation outlined above. On the other hand, we were faced with the practical problem that the identified variables were highly collinear, which led to multiple issues of multicollinearity when estimating regressions.

The next practical steps in Switzerland must now lie in determining how the identified structural differences among hospitals should be selected and combined to compensate hospitals for uncontrollable organizational and regional differences. For this, we suggest a regression-based approach, which selects a limited number of independent variables by building on the findings presented herein.

We have now also tried to clarify our motivation and the added value of our study in the manuscript. However, we deliberately avoided to go into too much detail about the current specific situation in Switzerland, because the publication is directed towards an international readership. 

We hope that we were able to clarify these points for you.

- Einkaufsgemeinschaft HSK. Benchmark SwissDRG Tarifjahr 2021. HSK; 2020.

- Widmer P, Trottmann M, Telser H. Das Fallpauschalenmodell: Leistungsbezogene Basispreise unter SwissDRG. Polynomics; 2015

2. Last, section 5 "discussion" is mostly a repetition of section 4 "results".

Yes, you are right, there are some repetitions in the discussion section. We have now deleted some of them to make it less repetitive. However, we believe that some other repetitions should remain, because they make our deliberations during the discussion easier to follow.

.

---

## [Editor Report · Decision Letter 1]

7 Feb 2022

Identification and assessment of a comprehensive set of structural factors associated with hospital costs in Switzerland

PONE-D-21-25093R1

Dear Dr. Havranek,

We’re pleased to inform you that your manuscript has been judged scientifically suitable for publication and will be formally accepted for publication once it meets all outstanding technical requirements.

Kind regards,

Matteo Lippi Bruni, PhD

Academic Editor

PLOS ONE
---

## [Editor Report · Acceptance letter]

8 Feb 2022

PONE-D-21-25093R1 

Identification and assessment of a comprehensive set of structural factors associated with hospital costs in Switzerland 

Dear Dr. Havranek:

I'm pleased to inform you that your manuscript has been deemed suitable for publication in PLOS ONE. Congratulations! Your manuscript is now with our production department. 

Kind regards, 

on behalf of

Dr. Matteo Lippi Bruni 

Academic Editor

PLOS ONE